# Live-cell single particle imaging reveals the role of RNA polymerase II in histone H2A.Z eviction

Anand Ranjan[1], Vu Q Nguyen[1], Sheng Liu[1], Jan Wisniewski[2†], Jee Min Kim[1], Xiaona Tang[1], Gaku Mizuguchi[1], Ejlal Elalaoui[1], Timothy J Nickels[1], Vivian Jou[1], Brian P English[2], Qinsi Zheng[2], Ed Luk[3], Luke D Lavis[2], Timothee Lionnet[4], Carl Wu[1,5]*

[1]Department of Biology, Johns Hopkins University, Baltimore, United States; [2]Janelia Research Campus, Howard Hughes Medical Institute, Ashburn, United States; [3]Department of Biochemistry and Cell Biology, Stony Brook University, Stony Brook, United States; [4]Institute of Systems Genetics, Langone Medical Center, New York University, New York, United States; [5]Department of Molecular Biology and Genetics, Johns Hopkins School of Medicine, Baltimore, United States

*For correspondence: wuc@jhu.edu

Present address: †Experimental Immunology Branch, National Cancer Institute, Bethesda, USA

Competing interests: The authors declare that no competing interests exist.

**Abstract** The H2A.Z histone variant, a genome-wide hallmark of permissive chromatin, is enriched near transcription start sites in all eukaryotes. H2A.Z is deposited by the SWR1 chromatin remodeler and evicted by unclear mechanisms. We tracked H2A.Z in living yeast at single-molecule resolution, and found that H2A.Z eviction is dependent on RNA Polymerase II (Pol II) and the Kin28/Cdk7 kinase, which phosphorylates Serine 5 of heptapeptide repeats on the carboxy-terminal domain of the largest Pol II subunit Rpb1. These findings link H2A.Z eviction to transcription initiation, promoter escape and early elongation activities of Pol II. Because passage of Pol II through +1 nucleosomes genome-wide would obligate H2A.Z turnover, we propose that global transcription at yeast promoters is responsible for eviction of H2A.Z. Such usage of yeast Pol II suggests a general mechanism coupling eukaryotic transcription to erasure of the H2A.Z epigenetic signal.

## Introduction

The H2A.Z variant of canonical histone H2A serves as a key chromatin constituent of the epigenome, providing a unique nucleosome architecture and molecular signature for eukaryotic gene transcription and other chromosome activities (*Weber and Henikoff, 2014*). H2A.Z is enriched at most promoters and enhancers genome-wide, and plays a role in establishing a permissive chromatin state for regulated transcription (*Weber and Henikoff, 2014*). H2A.Z is incorporated in nucleosomes flanking DNase hypersensitive, nucleosome-depleted regions (NDRs), especially at the so-called '+1 nucleosome' overlapping with or immediately downstream of the transcription start site (TSS) (*Albert et al., 2007*; *Weber and Henikoff, 2014*). The deposition of H2A.Z in budding yeast is catalyzed by the conserved SWR1 chromatin remodeling complex in an ATP-dependent reaction involving exchange of nucleosomal H2A-H2B for H2A.Z-H2B dimers (*Mizuguchi et al., 2004*).

Genome-wide studies have shown that compared to nucleosomes in the gene body, the +1 nucleosome undergoes higher turnover, which is not correlated with the level of mRNA transcription by Pol II (*Dion et al., 2007*; *Grimaldi et al., 2014*; *Rufiange et al., 2007*). Thus, the disruptive passage of Pol II through +1 nucleosomes during infrequent mRNA transcription is unlikely to account for H2A.Z eviction on a global scale. Biochemical studies have suggested that yeast H2A.Z eviction could be due to chromatin remodeling in reverse mediated by SWR1 itself (*Watanabe et al., 2013*)

or the related INO80 remodeler (*Papamichos-Chronakis et al., 2011*), but other biochemical studies found no supporting evidence (*Luk et al., 2010*; *Wang et al., 2016*). Alternatively, genome-wide assembly of the transcription pre-initiation complex (PIC) has been proposed to evict H2A.Z, but the key event in this multistep process remains elusive (*Tramantano et al., 2016*). To determine the dominant mechanism of H2A.Z turnover after incorporation, we took an independent approach using single-particle tracking that directly measures the levels of chromatin-free and bound H2A.Z in the physiological environment of living yeast cells, in wild type and conditional mutants for histone eviction.

Single-particle tracking (SPT) of fluorescently tagged proteins in live cells has emerged as a robust imaging technique to determine kinetic behaviors of protein factors (*Elf and Barkefors, 2019*; *Liu and Tjian, 2018*). For chromatin-interacting proteins, SPT is complementary to genome-wide chromatin immunoprecipitation-DNA sequencing technologies (ChIP-seq) without the general caveats of chemical fixation and chromatin manipulations. SPT directly measures the fast-diffusing, chromatin-free population as well as the quasi-immobile, chromatin-bound fraction tracking with macroscopic chromosome movements (*Liu and Tjian, 2018*; *Taddei and Gasser, 2012*). Here, we combine SPT in yeast with conditional depletion of candidate regulators to study the dynamic deposition and eviction of histone H2A.Z. Our results indicate a prominent role of RNA Polymerase II in the removal of H2A.Z from chromatin.

## Results

### Tracking single molecules of histones and remodeler SWR1

We fused the self-labeling HaloTag to H2A.Z, H2B, and Swr1 (the catalytic subunit of the SWR1 complex) for sole source expression under native promoter control and validated the function of these fusion constructs (*Figure 1—figure supplement 1A*). Yeast cultures were fluorescently labeled to saturation with Janelia Fluor 646 (*Grimm et al., 2015*; *Figure 1—figure supplement 1B,C*), and movies of single molecules were recorded at high temporal resolution (10 ms exposure) in live cells (*Rust et al., 2006*; *Shim et al., 2012*; *Videos 1–3*). Single molecule trajectories (n > 1000 and ≥6 frames for each trajectory) were obtained from over 50 yeast cells for each strain. The data are presented as histograms of particle frequency over the diffusion coefficient (log D) extracted from mean squared displacements (MSD) (*Figure 1A–D*, and methods). For a more robust quantitation of diffusive populations, we also applied a kinetic modeling approach ('Spot-On') based on single particle displacements (*Hansen et al., 2018*; *Figure 1E,F*). We performed Spot-On analysis on single-molecule trajectories (≥3 frames), cite Spot-On values for chromatin-bound and chromatin-free fractions in the text, and provide results from both Spot-On and MSD analyses in all figures.

The SPT profiles for H2A.Z and H2B were best fitted by a simple model comprised of two diffusive populations—a major, slow-diffusing chromatin-bound fraction (H2A.Z: 82%, H2B: 76%, average D: 0.03 $\mu m^2 s^{-1}$), and a minor, fast-diffusing chromatin-free fraction (H2A.Z: 1.18 $\mu m^2 s^{-1}$, H2B: 1.29 $\mu m^2 s^{-1}$) (*Figure 1E,F* and *Figure 1—figure supplement 2A,B,E*). Additional minor populations of H2A.Z and H2B with distinct diffusive values are not excluded. The fraction of chromatin-bound H2A.Z was consistent with a previous estimate by in vivo cross-linking (*Mohan et al., 2018*), and the D value of bound yeast H2B was also consistent with that of mammalian H2B (0.02 $\mu m^2 s^{-1}$) in a previous report (*Hansen et al., 2018*). The 'free' H2A.Z fraction represents soluble H2A.Z-H2B dimers biochemically associated with histone chaperones, in addition to a minor population in complex with the ~1 MDa SWR1 complex (*Luk et al., 2007*). We observed similar frequencies of chromatin-bound and free H2A.Z in cells growing synchronously after release from G1 arrest into S phase (*Figure 1—figure supplement 3*).

In contrast to behaviors of the histones, the SWR1 complex (Swr1-Halo subunit) showed more chromatin-free diffusion. In addition, deletion of Swc2, a key subunit involved in the recruitment of SWR1 to gene promoters (*Ranjan et al., 2013*), substantially reduced the chromatin-bound fraction from 47% to 21% (*Figure 1C,D,F*). (Our imaging regime captures both stable and transiently bound SWR1 in the slow-diffusing population; the remaining 21% of slow molecules for the *swc2Δ* mutant may be largely attributed to transient binding). With these validations, we proceeded to investigate regulators of H2A.Z dynamics, based on the fractional changes in chromatin-bound and free H2A.Z. Notably, while the aforementioned labeling of HaloTag was adequately conducted with the JF646

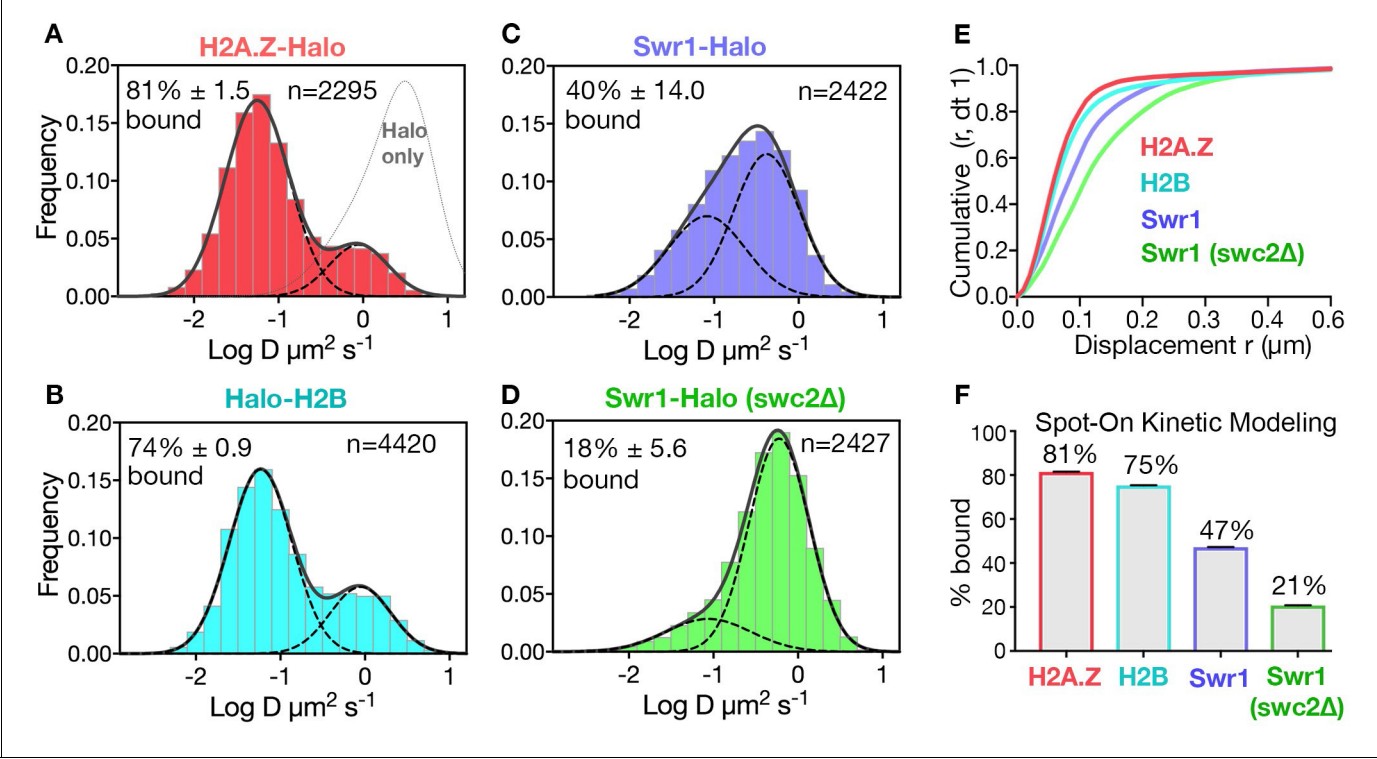

**Figure 1.** Diffusive behaviors of protein fusions to HaloTag (Halo) reveal chromatin-bound and free populations in live yeast. (**A, B**) Normalized histograms and two-component Gaussian fits for H2A.Z-Halo (**A**) and Halo-H2B (**B**) show the log diffusion coefficient distributions. The Gaussian fit for HaloTag is shown for reference ('Halo only' in A). (**C, D**) Normalized histograms and two-component Gaussian fits for Swr1-Halo in WT cells (**C**) and the *swc2Δ* mutant (**D**). Solid line: sum of two-component fit; dashed line: individual component. Percent value of the slow component along with Bootstrap resampling errors and the number of trajectories (n) are indicated. (**E**) Cumulative distribution functions (CDF) of 10 ms displacements. (**F**) Spot-On results with fitting errors showing fractions of chromatin-bound molecules derived from modeling CDFs over 10–50 ms intervals. All molecules tracked with JF646 dye except Halo only, which was imaged with JF552.

The online version of this article includes the following figure supplement(s) for figure 1:

**Figure supplement 1.** Cell growth, labeling and SPT analysis of Halo-tagged proteins.
**Figure supplement 2.** Spot-On kinetic modeling analysis.
**Figure supplement 3.** H2A.Z-Halo distribution in cell division cycle.

dye, a superior flurophore JF552 became available in the course of this work, prompting its use in subsequent experiments for improved signal to noise (*Zheng et al., 2019*; *Figure 1—figure supplement 1F*).

## Eviction of H2A.Z upon SWR1 depletion in live yeast

The steady-state chromatin occupancy for H2A.Z is a function of competing deposition and eviction pathways. To highlight H2A.Z eviction in live cells, we blocked the H2A.Z incorporation pathway at gene promoters by conditional 'anchor-away' (AA) depletion of the Swc5 subunit, which is not required for Swr1 recruitment (*Figure 2—figure supplement 1*), but essential for SWR1 activity (*Haruki et al., 2008*; *Sun and Luk, 2017*; *Tramantano et al., 2016*). In the AA system, rapamycin mediates heterodimerization of FRB and FKBP12 moieties fused to Swc5 and the ribosomal protein RPL13A, respectively (i.e. Swc5-FRB and RPL13A-FKBP12), thus depleting Swc5 from the nucleus along with pre-ribosomal subunit export (*Haruki et al., 2008*). The 'wild type' AA yeast strain, mutated for TOR1 and the FK506-binding protein, is physiologically resistant to rapamycin and displays normal growth phenotype and normal H2A.Z and Pol II distributions (*Tramantano et al., 2016*; *Wong et al., 2014*).

Upon Swc5 AA, we found the expected decrease of chromatin-bound H2A.Z from 79% to 49% (*Figure 2A–C* and rapamycin time course in *Figure 2—figure supplement 2*), consistent with ChIP-

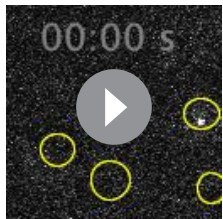

**Video 1.** H2A.Z-Halo single molecules imaged in wild type cells. Molecules were tracked with JF552 dye. Time is indicated on top in seconds and outline of nuclear regions are marked in yellow.
https://elifesciences.org/articles/55667#video1

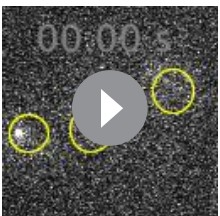

**Video 2.** Swr1-Halo single molecules imaged in wild type cells. Molecules were tracked with JF646 dye. Time is indicated on top in seconds and outline of nuclear regions are marked in yellow.
https://elifesciences.org/articles/55667#video2

seq results showing genome-wide reduction of H2A.Z at +1 nucleosomes under similar conditions (*Tramantano et al., 2016*). The remaining chromatin-bound H2A.Z may be due to residual H2A.Z at the +1 nucleosome or to histone chaperone-mediated H2A.Z deposition in nucleosomes over the entire genome, as suggested by in vivo cross-linking studies (*Mohan et al., 2018*; *Tramantano et al., 2016*). Our live-cell findings are thus consistent with the SWR1 requirement for promoter-specific H2A.Z deposition.

## Role of RNA Pol II in H2A.Z eviction

To identify H2A.Z eviction factors, we tested candidates that could mitigate the loss of chromatin-bound H2A.Z when both deposition and eviction factors were co-depleted in a double AA experiment. As the transcription PIC is constitutively enriched at the majority of NDRs (*Rhee and Pugh, 2012*) and has been causally linked to H2A.Z eviction (*Tramantano et al., 2016*), we first imaged the distribution of H2A.Z after nuclear depletion of both Swc5 and the Rpb1 catalytic subunit of Pol II. When Swc5 and Rpb1 are co-depleted by double AA, the chromatin-bound H2A.Z fraction increased (66%) relative to Swc5 AA alone (49%) (compare *Figure 3A* to *Figure 2C* and *Figure 3A* to *Figure 2D*). A complete restoration to the wild type level of H2A.Z is not anticipated because co-depletion of Swc5 also reduces H2A.Z deposition by SWR1 during anchor away. Fluorescence microscopy confirmed relocation of Swc5 to the cytoplasm in double AA cells, excluding inefficient nuclear depletion as a caveat (*Figure 3—figure supplement 1C,D*). These results indicate that Pol II indeed plays a major role in H2A.Z eviction. (Single AA of Rpb1 in rapamycin-treated cells showed a marginal increase from 84% to 87% of the bulk chromatin-bound H2A.Z over the untreated control [*Figure 3—figure supplement 2A–C*]).

## INO80 is not required for eviction of H2A.Z

To examine the role of the INO80 remodeler in H2A.Z eviction, we analyzed the H2A.Z distribution for Swc5 and Ino80 co-depletion by double AA and found no rise in bound H2A.Z compared to the single AA of Swc5 (compare *Figure 3A* to *Figure 2C* and *Figure 3C* to *Figure 2D*). (We observed no change in chromatin-bound H2A.Z for single AA of Ino80 (*Figure 3—figure supplement 2D–F*)). Taken together, we conclude that Pol II, but not the INO80 remodeler, has a major role in H2A.Z eviction. Contributions by other factors such as the ANP32E histone chaperone found in mammalian cells are not excluded (*Mao et al., 2014*; *Obri et al., 2014*).

## Kin28/CDK7 affects H2A.Z eviction

Transcription by Pol II is a complex process involving PIC assembly, Pol II initiation, promoter escape, productive elongation and termination (*Jonkers and Lis, 2015*; *Sainsbury et al., 2015*). Given that site-specific phosphorylation of the Rpb1 subunit of Pol II regulates the progression of transcription, targeted depletion of transcriptional kinases provides an opportunity to identify the step involved in H2A.Z eviction. A key post-initiation step involves Serine five phosphorylation (Ser5-P) of heptapeptide repeats on the C-terminal domain (CTD) of Pol II (Rpb1) (*Corden, 2013*; *Harlen and Churchman, 2017*). Ser5-P is catalyzed by the yeast Kin28/Cdk7 kinase, a component of the kinase module (Kin28-Ccl1-Tfb3) of TFIIH, and is linked to capping of nascent RNA, Pol II release from the Mediator

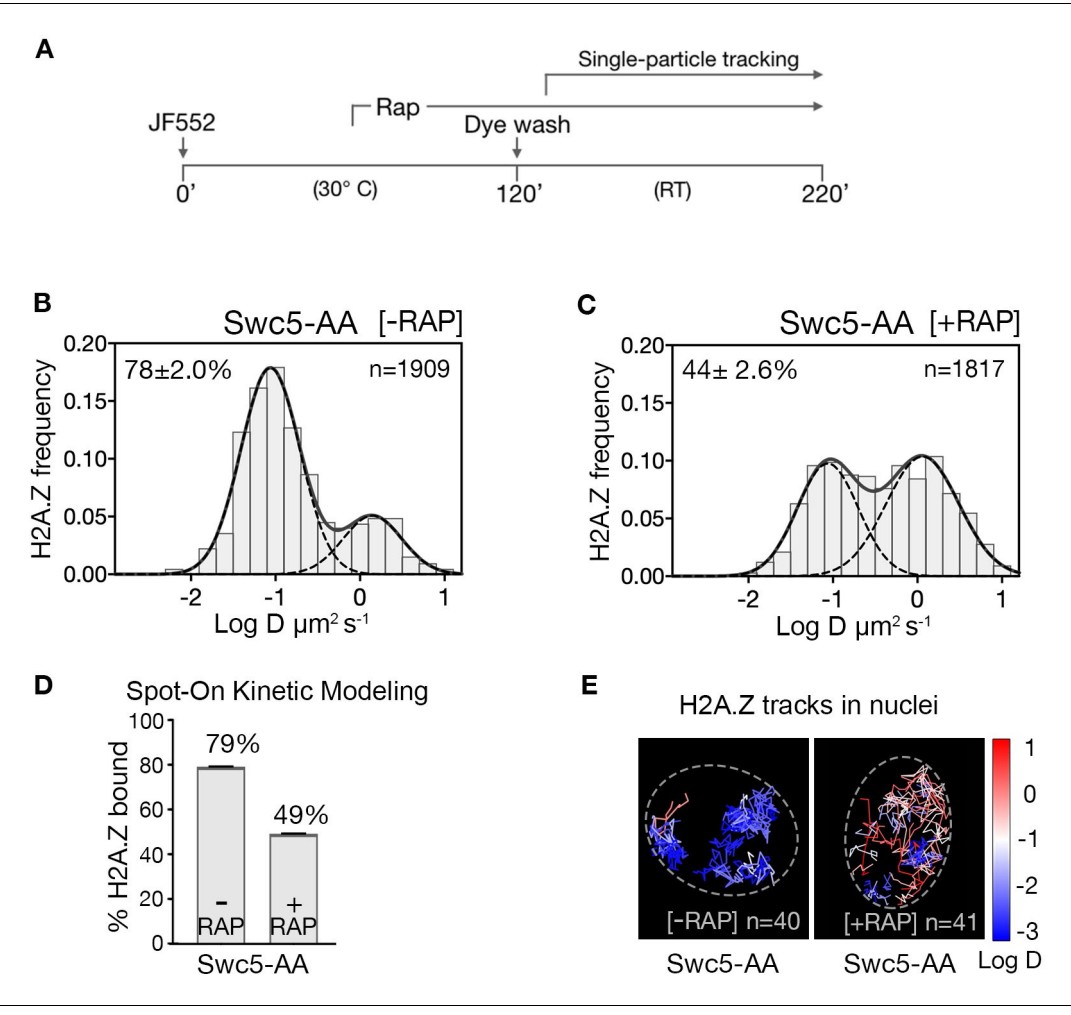

**Figure 2.** H2A.Z chromatin binding is substantially reduced upon abrogation of the deposition pathway by SWR1 inactivation. (A) Time course of H2A.Z-Halo labeling, rapamycin treatment and image acquisition in Swc5-AA cells. Rapamycin treatment for an hour before SPT, and imaging performed in continued presence of rapamycin. (B, C) Normalized histograms and two-component Gaussian fits for H2A.Z-Halo imaged in the Swc5-AA cells. Imaging data were acquired in absence of rapamycin (B) or presence of rapamycin (C). Spot-On results show that Swc5 depletion causes a reduction in chromatin-bound H2A.Z. (E) Overlay of tracks, color-coded according to log diffusion coefficients, obtained from representative nuclei. Number of tracks (n) is indicated for each nucleus. All molecules tracked with JF552 dye.

The online version of this article includes the following figure supplement(s) for figure 2:

**Figure supplement 1.** Swc5 is required post-recruitment for SWR1 activity.
**Figure supplement 2.** Reduction in chromatin-bound H2A.Z in a time course of Swc5-AA.

---

complex, promoter escape and early elongation. Recently, the Bur1/Cdk9 kinase was shown to phosphorylate the Rpb1 linker just upstream of CTD, at residues Thr 1471 and Ser 1493 (***Chun et al., 2019***), facilitating Pol II transition from early elongation to productive elongation. Furthermore, the Ctk1/Cdk12 kinase mediates Ser2 phosphorylation of the CTD associated with productive elongation through protein-coding regions (***Corden, 2013***; ***Harlen and Churchman, 2017***; ***Wong et al., 2014***). To investigate which phosphorylated state of Pol II is linked to H2A.Z eviction, we examined H2A.Z distributions in double AA cells conditionally deficient for Swc5 in combination with each of the three CTD kinases. Only Kin28 is required for H2A.Z eviction, as indicated by 65% chromatin-bound H2A.Z in the Kin28 and Swc5 double AA relative to 49% in the single AA of Swc5 (compare ***Figure 4B*** to ***Figure 2C*** and ***Figure 4F*** to ***Figure 2D***). Consistent with its role in H2A.Z eviction, depletion of Kin28 alone showed an increase in chromatin-bound H2A.Z, though marginal

(from 78% to 82%, *Figure 4—figure supplement 1A–C*). In contrast, double AA of Swc5 and the Bur1 kinase did not inhibit loss of chromatin-bound H2A.Z, nor did double AA of Swc5 and the Ctk1 kinase (*Figure 4C, D and F*).

## H2A.Z removal does not require RNA capping

In the wake of Pol II initiation, nascent RNA is co-transcriptionally capped by the sequential activity of three enzymes—Cet1, Ceg1 and Abd1—and is completed when RNA reaches ~100 nt (*Lidschreiber et al., 2013*). Capping of the 5' end of nascent RNA is initiated by the Cet1-Ceg1 complex, which recognizes the 5' triphosphate on the RNA and Ser5-P on the Pol II CTD (*Martinez-Rucobo et al., 2015*). To examine whether RNA capping or associated activities are required for H2A.Z eviction, we performed double AA of Swc5 and Cet1, and found no increase in chromatin-bound H2A.Z compared to single AA of Swc5 (compare *Figure 4E* to *Figure 2C* and *Figure 4F* to *Figure 2D*). Thus, H2A.Z eviction is not dependent on RNA capping. Likewise, we found no increase of chromatin-bound H2A.Z upon double AA of Swc5 and Rrp6, the 3'−5' exonuclease responsible for degradation of noncoding RNA (*Figure 4—figure supplement 1E,F*). Taken together, we conclude that an early stage of transcription elongation closely linked to Pol II CTD Ser5 phosphorylation by Kin28 is required for robust eviction of chromatin-bound H2A.Z.

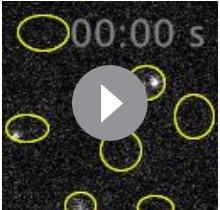

**Video 3.** H2A.Z-Halo single molecules imaged after anchor-away of swc5. Molecules were tracked with JF552 dye. Time is indicated on top in seconds and outline of nuclear regions are marked in yellow. https://elifesciences.org/articles/55667#video3

## Discussion

Transcription of most yeast genes is infrequent and nucleosome turnover along gene bodies is low, but the +1 nucleosome constitutively turns over at a 3-fold higher rate (*Dion et al., 2007*; *Grimaldi et al., 2014*; *Yen et al., 2013*). Similarly, H2A.Z is constitutively displaced from +1 nucleosomes for both active and rarely transcribed genes, on a timescale of <15 min (*Tramantano et al., 2016*). The live-cell SPT approach shows that Pol II rather than the INO80 sub-family of remodelers plays a key role in H2A.Z eviction. The INO80 sub-family of remodelers comprise of SWR1 and

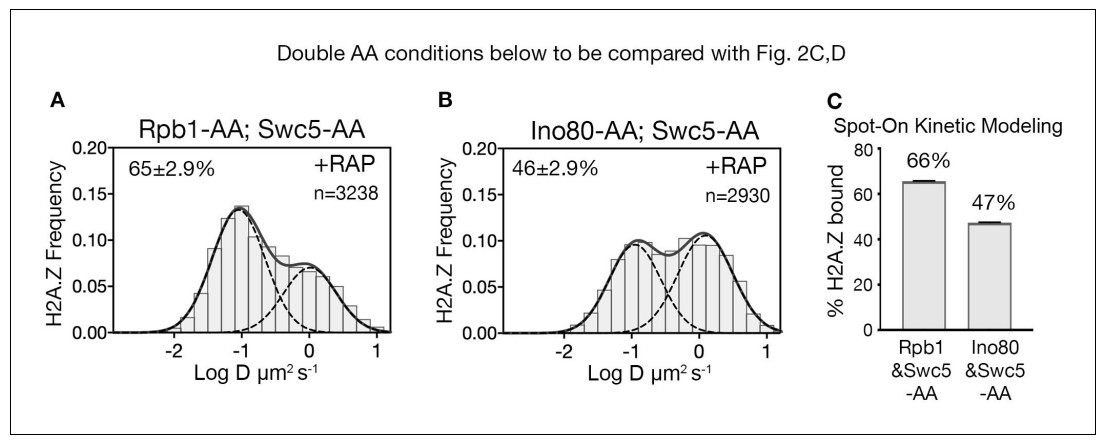

**Figure 3.** RNA polymerase II is critical for H2A.Z eviction. (**A**) Normalized histograms and two-component Gaussian fits for H2A.Z-Halo imaged in cells co-depleted for Rpb1 and Swc5. (**B**) H2A.Z-Halo distributions in cells co-depleted for Ino80 and Swc5. (**C**) Spot-On results showing co-depletion of Rpb1 along with Swc5 inhibits H2A.Z eviction. All molecules tracked with JF552 dye.

The online version of this article includes the following figure supplement(s) for figure 3:

**Figure supplement 1.** Efficient nuclear depletion of Swc5 in double anchor-away (*SWC5-FRB; RPB1-FRB*) strain.
**Figure supplement 2.** H2A.Z diffusion histograms in cells for single AA of Rpb1 and Ino80.

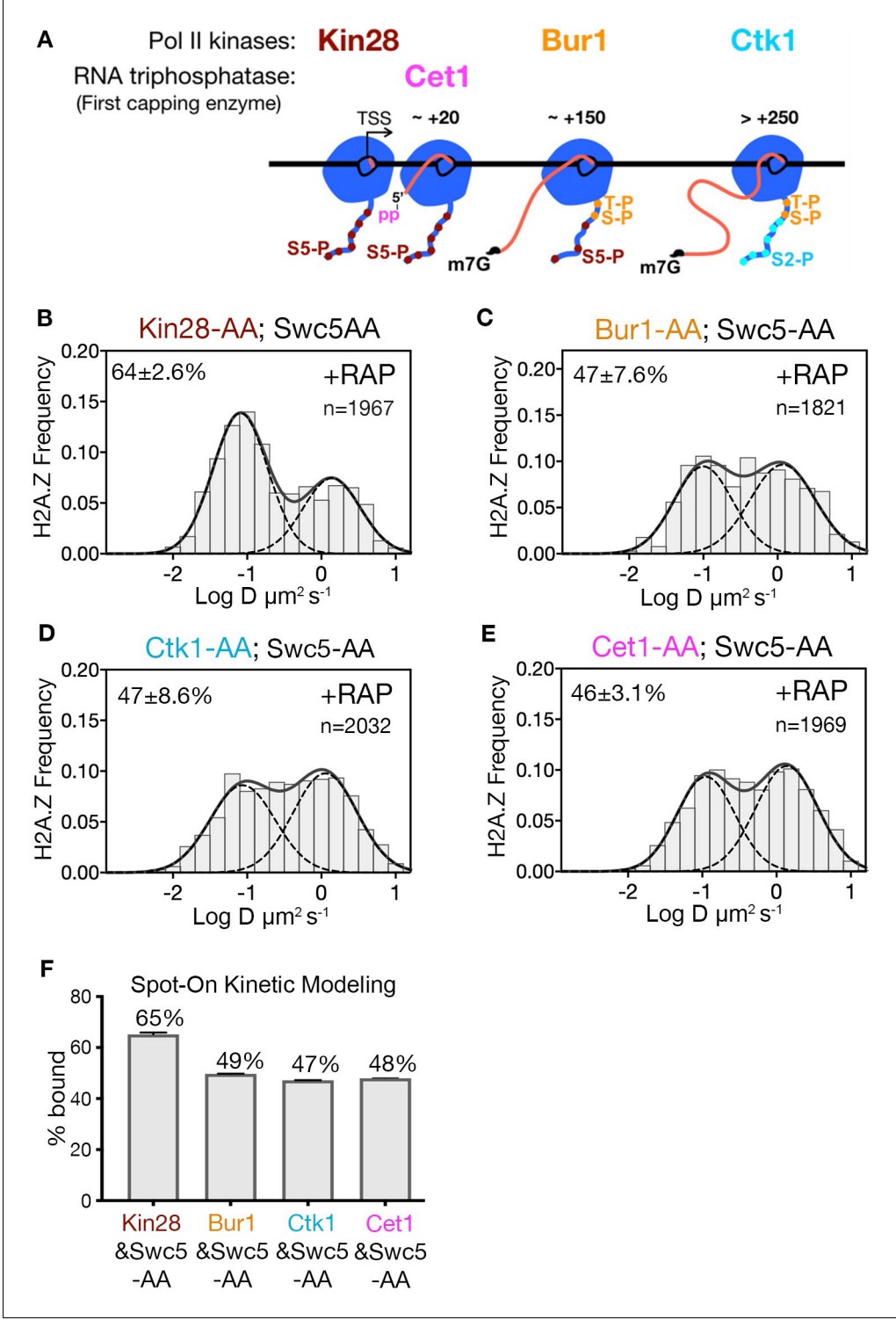

**Figure 4.** Kin28 phosphorylation of RNA polymerase II CTD is critical for H2A.Z eviction. (**A**) Schematic representation shows the three Pol II kinases Kin28, Bur1 and Ctk1 recruited at initiation, early-elongation and elongation phases respectively of Pol II and corresponding phosphorylation of indicated Rpb1 CTD sites. Set1 is the first of the three RNA capping enzymes; it removes γ-phosphate from the RNA 5'end to generate 5' diphosphate. (**B, C, D, E**) Normalized histograms and two-component Gaussian fits for H2A.Z-Halo imaged in cells co-depleted for Swc5 along with Kin28 (**B**), Bur1 (**C**), Ctk1 (**D**) and Cet1 (**E**). (**F**) Spot-On results show Kin28 is required to evict H2A.Z. All molecules tracked with JF552 dye.

*Figure 4 continued on next page*

*Figure 4 continued*

The online version of this article includes the following figure supplement(s) for figure 4:

**Figure supplement 1.** H2A.Z diffusion histograms in cells after single depletion of Kin28.

INO80 remodeling complexes, with similarities in the catalytic split-ATPase domain and four shared subunits (Rvb1, Rvb2, Act1 and Arp4). INO80 has been previously suggested to mediate the reversal of SWR1-catalyzed H2A.Z deposition (*Papamichos-Chronakis et al., 2011*). However, in addition to this study, there are several publications reporting that INO80 displays no detectable H2A.Z eviction from chromatin in either yeast or mammalian cells (*Au-Yeung and Horvath, 2018*; *Tramantano et al., 2016*; *Jeronimo et al., 2015*). Although evidently uninvolved in H2A.Z eviction, the INO80 complex is known to mediate nucleosome repositioning in vitro and in vivo (*Kubik et al., 2019*; *Shen et al., 2003*).

Our results confirm the previous finding that the Pol II PIC plays an important role in H2A.Z eviction. Furthermore, the observed dependence on Kin28/Cdk7 kinase for robust H2A.Z displacement suggests that PIC assembly per se, that is the recruitment of general transcription factors and Pol II to promoter, is not sufficient for H2A.Z eviction but Kin28-dependent phosphorylation of Ser5 of the CTD heptapeptide repeats is important (*Figure 5*, box). A role for Kin28 in Ssl2-facilitated TSS scanning by Pol II is not excluded for H2A.Z eviction, although it has been shown that depletion of Kin28 by AA does not alter TSS usage (*Murakami et al., 2015*).

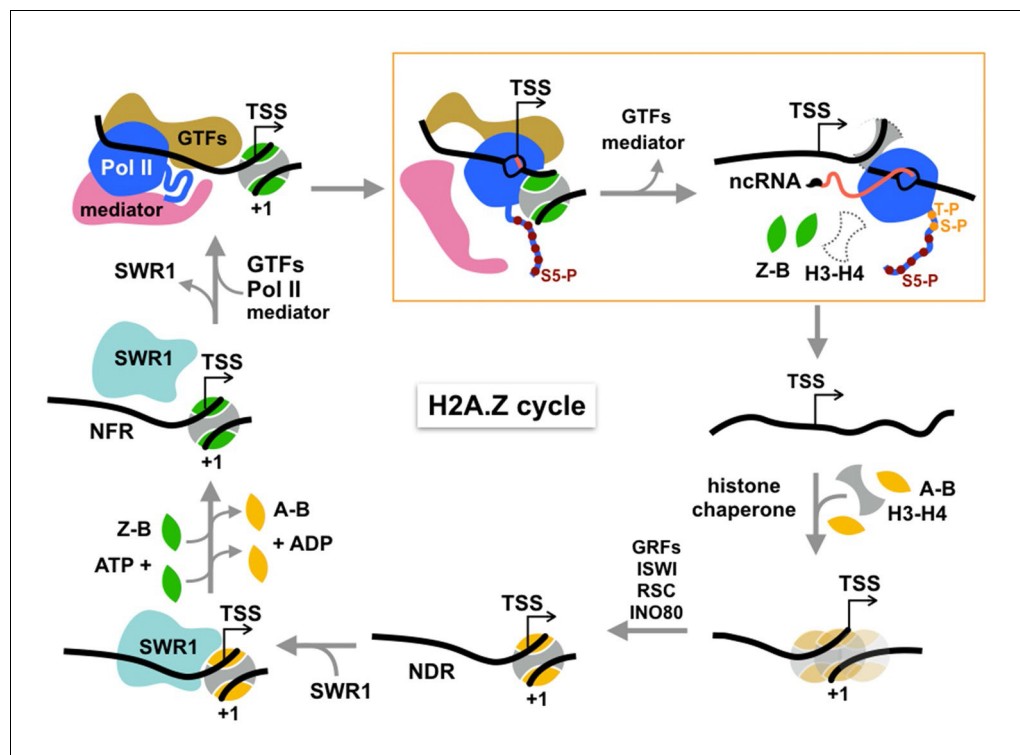

**Figure 5.** Cycle of H2A.Z eviction and deposition. RNA polymerase II assembled genome-wide in the PIC and Rpb1 CTD Ser5 phosphorylated by Kin28 constitutively transcribes short noncoding RNAs (with m7G cap) and evicts H2A.Z-H2B dimers from the +1 nucleosome prior to termination. H2A.Z eviction should also occur in the course of mRNA transcription. Additional factors may be necessary for displacement of H3-H4 tetramer. The directional arrow indicates the annotated transcription start site. The gap is filled by histone chaperone-mediated deposition of canonical histones to reform an H2A-containing +1 nucleosome, which is positioned by chromatin remodelers and sequence-specific transcription factors, maintaining the NDR. This recruits SWR1 which is activated upon recognition of H2A-nucleosome and H2A.Z-H2B dimer substrates to activate one or two rounds of H2A.Z deposition. See text for discussion.

Inhibition of H2A.Z eviction upon depletion of Kin28/Cdk7 kinase, but not Bur1/Cdk9 or Ctk1/Cdk12 kinases narrows the relevant state of Pol II to early elongation after promoter escape, but not to productive elongation. (We note that the PIC remains largely intact upon Kin28/Cdk7 depletion, as shown by accumulation of TFIID, Mediator and Pol II at gene promoters [*Knoll et al., 2019*; *Wong et al., 2014*]).The exclusive dependence on Kin28/Cdk7 is further underscored by no reduction of H2A.Z eviction on depletion of Cet1, the 5′ RNA capping enzyme, or depletion of Rrp6, the 3′−5′ exonuclease for noncoding RNA degradation. Thus, H2A.Z eviction is independent of RNA modifying and metabolizing activities just downstream of CTD Ser5 phosphorylation.

Transcriptional elongation by Pol II is known to cause displacement of nucleosomal histones in biochemical assays (*Lorch et al., 1987*), providing a mechanism for H2A.Z turnover at the +1 nucleosome in the process of transcription through protein-coding regions. We propose a similar mechanism for genes that do not engage in productive transcription of mRNA, but exhibit genome-wide, constitutive transcription of noncoding RNAs which are prematurely terminated by the Nrd1-Nab3-Sen1 pathway in budding yeast (*Schaughency et al., 2014*). The early elongation activity of Pol II would dislodge H2A.Z-H2B dimers from the histone octamer of the +1 nucleosome. Displacement of the more stably bound H3-H4 tetramer likely requires assistance from histone chaperones and/or other remodelers (*Figure 5*).

After displacement of core histones, reassembly of a canonical nucleosome on gapped chromatin should occur, mediated by the mass action of the predominating H2A-H2B histone pool and histone chaperones, nucleosome positions being reset by chromatin remodelers such as ISWI, RSC, and INO80 (*Lai and Pugh, 2017  Figure 5*). Maintenance of a NDR of sufficient length (>60 bp DNA) by remodelers and subsequent histone acetylation recruits SWR1 to canonical +1 nucleosomes, the essential substrate for SWR1 (*Ranjan et al., 2013*). Stimulation of the catalytic Swr1 ATPase by nucleosome and H2A.Z-H2B dimer substrates then triggers histone dimer exchange (*Luk et al., 2010*; *Ranjan et al., 2013*), completing the cycle of H2A.Z/H2A replacement (*Figure 5*).

We envision that H2A.Z eviction is coupled to transcription not only from protein-coding genes transcribed by Pol II but also ribosomal, 5S and tRNA genes transcribed by Pol I and Pol III. Because H2A.Z eviction is not correlated with mRNA transcription by Pol II (*Tramantano et al., 2016*), the constitutive global transcription of noncoding RNA by Pol II is additionally coupled to H2A.Z eviction. There is substantial evidence for low-level, heterogenous transcripts of several hundred nucleotides, initiating from multiple start-sites within yeast NDRs (*Pelechano et al., 2013*). For budding yeast, these noncoding RNA transcripts evidently result from Pol II initiation without substantial pausing (*Booth et al., 2016*). At metazoan promoters, turnover of H2A.Z enriched in +1 nucleosomes may be similarly coupled to transcription in the process of Pol II pausing and release (*Tome et al., 2018*). Likewise at metazoan enhancers, infrequent Pol II transcription of eRNAs (*Tippens et al., 2018*) could be responsible for eviction of H2A.Z, representing erasure of a permissive histone variant mark on the epigenome. Much remains to be learned about the functional significance of this process and its relationship to productive mRNA transcription, presenting an outstanding problem for future studies of chromatin dynamics in eukaryotic gene regulation.

# Materials and methods

## Yeast strains and plasmids

The plasmid for HaloTag (Halo) fusions was generated by cloning HaloTag (Promega) in the pBluescript SK (-) vector followed by insertion of a KanMx cassette (Kanamycin) or NatMx cassette (Nourseothricin), following standard procedures (*Gelbart et al., 2001*). PCR amplification and standard yeast transformation methods were used for tagging the protein of interest at the C-terminus, with a serine-glycine $(SG_4)_2$ linker to HaloTag.

For Halo-H2B, plasmid HTA1-SNAP-HTB1 (pEL458, gift from Ed Luk) was modified to replace the SNAP coding sequence with HaloTag, with a four amino acid $GA_3$ linker between HaloTag and the N-terminus of H2B. The plasmid expressing Halo-H2B was introduced in the FY406 strain (gift from Fred Winston) by the plasmid shuffle procedure (*Hirschhorn et al., 1995*). The endogenous H2B promoter drives expression of Halo-H2B as the sole gene copy in cells.

Free HaloTag was fused at the N-terminus to a bipartite SV40 NLS (KRTADGSEFESPKKKRKV, where two clusters of basic residues are underlined) (*Hodel et al., 2006*) and expressed from the

pRS416 vector. Plasmid pAC1056 expressing BPSV40 NLS-GFP (gift from Anita Corbett) was modified for free Halo expression.

Strains and plasmids used for anchor-away studies were obtained from Euroscarf.

The transporter gene PDR5 was deleted in all strains for retention of HTL-dye conjugate in live yeast cells. Strain genotypes are listed in *Supplementary file 1*.

### Flow cytometry analysis

Cells were fixed by adding two volumes of 100% ethanol and stored for one hour at 4°C. Cells were washed with 50 mM Tris-HCl (pH 7.5) buffer and digested with RNase (1 mg/ml) and RNase A (0.2 mg/ml) overnight at 37°C on a rotator. Proteins were digested with Proteinase K (1 µg/µl) at 50°C for 30 min. Cells were stained with 2 µM SYTOX (Tris buffer) at 4°C for 4 hr and sonicated on Diagenode Biorupter 300 for 10 s at high setting. Cells were scanned on LSR II FACS instrument.

### Cell culture and labeling

Cells were grown and imaged in CSM media (Complete Supplement Mixture) supplemented with 40 mg/L adenine hemisulfate. The JF646-HaloTag ligand was synthesized as previously described (*Grimm et al., 2015*). The new JF552 dye has a higher signal to noise ratio and is more photostable than JF646. The JF552 dye is a modification of JF549, with similar brightness, but enhanced cell permeability that allows its use for SPT in yeast (*Zheng et al., 2019*). For in vivo labeling, early log phase cells (O.D$_{600}$ 0.2) were labeled with JF-HaloTag ligand (10 nM for JF646 and 20 nM for JF552) for two hours at 30 degrees in suspension culture. Cells were washed four times with CSM to remove free dye.

Prior to use, 0.17 mm coverslips (Ø 25 mm, Electron Microscopy Services) were flamed to remove punctuated surface auto-fluorescence and to suppress dye binding, and coated with Concanvalin A (2 mg/ml) for 30 min at room temperature, and air-dried for one hour. Coverslips were assembled in a Ø 35 mm Attofluor chamber (Invitrogen). A 1 ml cell suspension was immobilized for 10 min and live cells were imaged in CSM media at room temperature. For anchor away experiments, rapamycin (1 µg/ml) was added one hour prior to imaging, and cells were imaged in the presence of rapamycin. Two biological replicates were performed for each experimental condition.

### Cell cycle synchronization

Cells were synchronized in G1 by adding α factor for 2 hr (3 µg/ml at 0 min and additional 2 and 1 µg/ml at 60 and 90 min respectively). High autofluorescence did not allow SMT in presence of α factor, which was removed by replacing culture medium. Cells released from G1 at room temperature took 40 min to enter S phase. Both for Pre-S and S phase SMT, cells were stained and synchronized in suspension culture and immobilized right before SMT.

### Wide-field single molecule imaging with epi-Illumination microscope

Single-molecule imaging was conducted on a Zeiss Observer Z1 microscope with a Zeiss Plan-Apochromat 150X/1.35 glycerin-immersion objective. Cells of interest were identified under infrared illumination (750 nm, 10 nm FWHM) using a near IR-CCD camera (IDS UI-3370CP-NIR-GL) and Semrock 743 nm/25 nm FWHM filter. A 555 nm (Crystalaser) or 637 nm (Vortran) laser was used for dye excitation, typically at 100 mW total power (TTL pulsed). All laser beams were spectrally filtered and combined using a custom beam combiner (by J.W., details available upon request). A Semrock FF01-750/SP filter was included at the output to remove any residual near infrared emission from lasers. Combined laser beams were collimated into a 2m-long Qioptic fiber (kineFLEX-P-2-S-405.640–0.7-APC-P2) with output through a 12 mm EFL reflective collimator (Thorlabs). The resulting Ø6mm Gaussian beam was introduced into the back port of the microscope. The following cubes were utilized in the microscope turret to direct excitation light towards the sample and filter fluorescence: 1) for JF646 - 648 beamsplitter and 676/29 nm filter, 2) for JF552 - 561 beamsplitter and 612/69 nm filter. Images were acquired with a Hamamatsu C9100-13 back-illuminated EM-CCD camera through additional FF01-750/SP and NF03-405/488/561/635E quad-notch filters. The camera was operated at −80°C with a typical EM gain of 1200 and directly controlled by laser emission via the TTL signal.

## Image acquisition

Images were obtained using either 637 nm laser (JF646) or 555 nm laser (JF552), of excitation intensity ~1KW/cm$^2$ and for each field of view ~7000 frames were captured. Single molecules were tracked using DiaTrack Version 3.05 software, with the following settings; remove blur 0.1, remove dim 70–100, maximum jump six pixels. Single molecule images were collected after pre-bleach of initial intense fluorescence (glow). While imaging with JF646, a 405 nm laser excitation (1–10 mW/cm$^2$, TTL pulses 2–5 ms per frame) was triggered to maintain single fluorophore detection density. Immobilized cells in CSM media were imaged over a 90 min imaging session.

## Analysis of single-molecule images

Movies with two dimensional single molecule data were analyzed by DiaTrack Version 3.05 (*Vallotton and Olivier, 2013*), which determines the precise position of single molecules by Gaussian intensity fitting and assembles particle trajectories over multiple frames. In Diatrack remove blur was set to 0.1, remove dim set at 70 and max jump set at five pixels, where each pixel was 107 nm. Trajectory data exported from Diatrack was further analyzed by a custom computational package 'Sojourner' (by S.L.). The package is available on Github (https://rdrr.io/github/sheng-liu/sojourner/). The Mean Squared Displacement (MSD) was calculated for all trajectories six frames or longer. Diffusion coefficients for individual molecules were calculated by unconstrained linear fit ($R_2 > 0.8$) of the MSD values computed for time lags ranging from 2 dt to 5 dt, where dt = 10 ms is the time interval between frames, and slope of linear fit was divided by 4 (pre-factor for 2-dimensional Brownian motion) (*Qian et al., 1991*). The histogram of log converted diffusion coefficients was fitted with double gaussian function from the 'mixtools' package (*Benaglia et al., 2009*) to estimate the fraction of chromatin-bound molecules (mean range between 0.050–0.112 $\mu m^2\ s^{-1}$). Standard error on the mean of each gaussian fit parameter was estimated using a bootstrap resampling approach (*Efron, 1979*).

The Spot-On analysis was performed on trajectories three frames or longer using the web-interface https://spoton.berkeley.edu/ (*Hansen et al., 2018*). The bound fractions and diffusion coefficients were extracted from the CDF of observed displacements over different time intervals. For Brownian motion in two dimensions, the probability that a particle starting from origin will be found within a circle of radius r at time interval $\Delta\tau$ is given as follows.

$$P(r, \tau) = \frac{r}{2D\tau} e^{\frac{-r^2}{4D\tau}}$$

where D is diffusion coefficient. In Spot-On, the cumulative displacement histograms were fitted with a 2-state model.

$$p(r, \tau) = F_1 \frac{r}{2(D_1\tau + \sigma^2)} e^{\frac{-r^2}{4(D_1\tau + \sigma^2)}} + Z_{CORR}(\tau, Z, D_2) F_2 \frac{r}{2(D_1\tau + \sigma^2)} e^{\frac{-r^2}{4(D_1\tau + \sigma^2)}}$$

where F1 and F2 are bound and free fractions, $\sigma$ is single molecule localization error, D1 and D2 are diffusion coefficients of bound and free fractions, and $Z_{CORR}$ is correction factor for fast molecules moving out of axial detection range (*Hansen et al., 2018*). The axial detection range for JF646 on our setup is 650 nm. The following settings were used on the Spot-On web interface: bin width 0.01, number of time points 6, jumps to consider 4, use entire trajectories-No, Max jump ($\mu$m) 1.2. For model fitting the following parameters were selected: D$_{bound}$ ($\mu m^2$/s) min 0.001 max 0.1, D$_{free}$ ($\mu m^2$/s) min 0.15 max 5, F$_{bound}$ min 0 max 1, Localization error ($\mu$m)- Fit from data-Yes min 0.01 max 0.1, dZ ($\mu$m) 0.65 for JF646 and dZ 0.6 for JF552, Use Z correction- Yes, Model Fit CDF, Iterations 3.

## Acknowledgements

This work is dedicated to the memory of Maxime Dahan, former project leader of the HHMI-Janelia Transcription Imaging Consortium. We thank Anita Corbett for reagents, Zhe Liu, Brian Mehl, Aseem Ansari and Herve Rouault for discussions, Felix Wu for image processing, Anders Hansen, Maxime Woringer, and Xavier Darzacq for consultation on the Spot-On program, Prashant Mishra and Munira Basrai for assistance with FACS analysis, and James Brandt and Yumi Kim for deconvolution microscopy, Debbie Wei for making yeast strains, Sun Jay Yoo and Taibo Li for computational assistance.

The study was supported by HHMI-Janelia Transcription Imaging Consortium funding to CW, TL, and LL, the Damon Runyon Cancer Research Foundation (V.N.), the Johns Hopkins Bloomberg Distinguished Professorship (CW), a grant to EL from the National Institutes of Health (GM104111), a grant to TL from National Institutes of Health (GM127538), and a grant to CW from the National Institutes of Health (GM125831).

## Additional information

### Funding

| Funder | Grant reference number | Author |
| --- | --- | --- |
| National Institutes of Health | GM125831 | Carl Wu |
| National Institutes of Health | GM127538 | Timothee Lionnet |
| National Institutes of Health | GM104111 | Ed Luk |
| HHMI | Transcription Imaging Consortium, Janelia Research Campus | Luke D Lavis Timothee Lionnet Carl Wu |
| Damon Runyon Cancer Research Foundation | | Vu Q Nguyen |
| Johns Hopkins University | Bloomberg Distinguished Professorship | Carl Wu |

The funders had no role in study design, data collection and interpretation, or the decision to submit the work for publication.

### Author contributions

Anand Ranjan, Conceptualization, Resources, Data curation, Software, Formal analysis, Supervision, Validation, Investigation, Visualization, Methodology, Writing - original draft, Writing - review and editing; Vu Q Nguyen, Resources, Data curation, Formal analysis, Investigation, Visualization, Writing - review and editing; Sheng Liu, Xiaona Tang, Software; Jan Wisniewski, Conceptualization, Resources, Supervision, Methodology, Writing - review and editing; Jee Min Kim, Data curation, Formal analysis; Gaku Mizuguchi, Qinsi Zheng, Luke D Lavis, Resources; Ejlal Elalaoui, Timothy J Nickels, Vivian Jou, Data curation; Brian P English, Methodology; Ed Luk, Resources, Writing - review and editing; Timothee Lionnet, Methodology, Writing - review and editing; Carl Wu, Conceptualization, Supervision, Funding acquisition, Writing - original draft, Project administration, Writing - review and editing

### Author ORCIDs

Anand Ranjan (iD) https://orcid.org/0000-0001-6071-6017
Brian P English (iD) http://orcid.org/0000-0002-4037-6294
Ed Luk (iD) http://orcid.org/0000-0002-6619-2258
Carl Wu (iD) https://orcid.org/0000-0001-6933-5763

### Decision letter and Author response

Decision letter https://doi.org/10.7554/eLife.55667.sa1
Author response https://doi.org/10.7554/eLife.55667.sa2

## Additional files

### Supplementary files

- Supplementary file 1. List of strains used in this study.
- Supplementary file 2. List of results from MSD and Spot-On analysis.
- Transparent reporting form

## Data availability

Imaging data have been deposited at Dryad DOI: https://doi.org/10.5061/dryad.43cp80c.

The following dataset was generated:

| Author(s) | Year | Dataset title | Dataset URL | Database and Identifier |
|---|---|---|---|---|
| Carl Wu | 2020 | Live-cell single particle imaging reveals the role of RNA polymerase II in histone H2A.Z eviction | https://doi.org/10.5061/dryad.43cp80c | Dryad Digital Repository, 10.5061/dryad.43cp80c |

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
