## [Decision Letter]

**Acceptance summary:**

Your work makes a significant addition to existing models of how histone variant deposition and Pol II transcription are linked.

**Decision letter after peer review:**

[Editors’ note: the authors submitted for reconsideration following the decision after peer review. What follows is the decision letter after the first round of review.]

Thank you for submitting your work entitled "Live-cell single particle imaging reveals the role of RNA polymerase II in histone H2A.Z eviction" for consideration by *eLife*. Your article has been reviewed by three peer reviewers, and the evaluation has been overseen by a Reviewing Editor and a Senior Editor. The following individual involved in review of your submission has agreed to reveal their identity: John T Lis (Reviewer #3).

As you will see from the individual reviews attached below all the reviewers found that the single-molecule experiments were carefully carried out and that the data is of high quality. However, in the discussion amongst the reviewers it was also agreed that the findings do not go sufficiently beyond the work in Tramantano et al., 2016. The reviewers felt that if additional experiments can be carried out to identify where in the transcription cycle H2A.Z is evicted, then the results would contribute both important and novel insights. These experiments as suggested by reviewers 2 and 3, could involve looking at RRP6 and XRN1 mutants and inhibitors of Cdk9. Such experiments would not only provide an important technologically distinct confirmation of previous conclusions from Tramantano et al., 2016 but provide sufficient new mechanistic insight to pass the bar for *eLife* publication.

As it is unclear how long it would take to acquire the necessary data, the reviewers concluded that the current manuscript should be rejected. We would however welcome a resubmission should additional data become available that addresses the reviewer concerns. Please feel free to communicate with us if you would like any clarification.

*Reviewer #1:*The histone variant H2AZ is often present near transcription start sites in yeast. How the steady state levels of this variant are maintained and the consequences of regulating H2AZ dynamics are areas of active study. While some in vitro studies have suggested that IN080 plays a role in evicting H2AZ, others have not. Further, previous work using ChIP-based methods (Tramantano et al., 2016) has indicated that rather than INO80, the Pol II transcription machinery plays a major role in evicting H2AZ histones from the +1 position at transcription start sites. Here the authors use live cell single particle tracking microscopy to directly study the effects of depleting different transcription regulators on H2AZ dynamics. Using the anchor-away (AA) method they find that H2AZ retention on chromatin (as assessed by the slower diffusing fraction) is decreased upon Swc5-AA and restored by simultaneous Rpb1-AA. In contrast, Ino80-AA does not substantially restore the H2AZ chromatin bound levels that are depleted by Swc5-AA. They further find that inhibition of Kin28 also restores a substantial portion chromatin bound H2AZ that is lost upon Swc5-AA. The authors interpret these results to suggest that H2AZ eviction is driven by the early elongating Pol II and therefore could reflect the consequences of nascent transcription of non-coding RNAs.

Overall the studies are technically of high quality and the experiments are carried out in a well-controlled manner. The main conclusion that H2AZ eviction depends on both Pol II and its Ser5 phosphorylated state would be of interest to the chromatin community. However, as is, the work does not go substantially beyond the insights obtained from Tramantano et al. At the same time, the hypothesis of non-coding RNA transcription driving H2AZ turnover is attractive and novel. If the authors could provide some test of this hypothesis, this would substantially raise the significance of the work.

Reviewer #2:

The manuscript described the role of Pol2 in H2A.Z dynamics by using single-molecule tracking. Authors measure H2A.Z bound fractions in various depletion conditions, concluding that H2A.Z eviction is dependent on Pol2-CTD serine-5p rather than INO80 complex. Although single-molecule tracking for H2A.Z in the various conditions is informative to the field, the biological significance seems to be a large repetition of the results in Tramantano, 2016, with different aspects of technique. It would be nice if authors could more clearly explain the novelty of their work and how it extends beyond Tramantano, 2016.

We think the Tramantano, 2016 paper is pretty tightly controlled and well-executed, even though its findings are largely correlative. Their correlations tell them that both Pol II and PIC are important for H2A.Z turnover at the +1 nucleosome, but the mechanism is unknown and which part of the PIC is necessary is unclear. So the major contribution from this paper is that a defective TFIIH can restore H2A.Z fraction bound to near WT levels in a Swc5-depleted background. Going back to the paper from which the Wu lab got the Kin28is mutant puts a question mark on their proposed mechanism of H2A.Z turnover. They found that in Kin28is cells treated with CMK, Pol II accumulates at the +2 nucleosome. So the Wu hypothesis that it is passage of Pol II through the +1 nucleosome that displaces H2A.Z doesn't make sense. It would be nice to have the authors comment on this and help clarifying their point.

-The key conclusion in the manuscript is that Pol2-CTD-S5p associates with H2A.Z eviction based on the kin28is model. However, it is problematic to disrupt the TFIIH helicase activity for this purpose, in which this would have nothing to do with phosphorylation but shutdown entire transcription.

-Authors used the time course with dye staining before anchoring protein away. This resulted in the possibility that a portion of SPT measurements might be derived from the "WT" condition. Could authors also demonstrate a time course with dye staining after anchoring target away, or staining with different dyes before and after adding rapamycin?

-Authors did not have any data supporting the role of ncRNA here. Authors need to provide the evidence by measuring H2A.Z dynamics in RRP6 or XRN1 mutants.*Reviewer #3:*

This paper from Carl Wu's group evaluates the mechanisms in vivo that lead to rapid exchange dynamics of the H2A.Z histone variant, which is known to occupy nucleosomes adjacent to active or activatable promoters. The H2A.Z variant is highly conserved across species and is known to be a critical: mutants produce slow growth phenotypes in yeast and are lethal in higher organisms. Understanding H2Az is important to evaluate of mechanisms of interplay of chromatin and promoters during gene expression.

This paper begins by emphasizing that the conserved SWR1 chromatin remodeling complex is responsible for exchanging nucleosomal H2A-H2B for H2A.Z-H2B dimers onto the +1 nucleosome in budding yeast. The +1 nucleosome undergoes much higher turnover than other nucleosomes and the mechanism by which H2A.Z is evicted and replaced by H2A had not been resolved. The authors point out that H2A.Z eviction could be due to chromatin remodeling in reverse mediated by SWR1 itself or the related INO80 remodeler, but a study (Wang et al., 2016) found no supporting evidence for either model. Pol II activity itself is an attractive candidate for H2A.Z eviction, as Pol II transcription can disrupt nucleosome structure, but the amount of H2A.Z eviction from genes does not correlate with the level of mRNA accumulation.

To assess what is the dominant mechanism of H2A.Z turnover after incorporation, the authors used single particle tracking (SPT) to directly measure the levels of chromatin-free and bound H2A.Z in living yeast cells, in WT and conditional mutants (or in cells treated with a specific Cdk7 inhibitor) of candidates/processes that are hypothesized to be involved in H2A.Z eviction. "SPT measures the fast diffusing, chromatin-free population as well as the quasi-immobile, chromatin-bound fraction tracking with macroscopic chromosome movements". From these studies, the authors conclude that "H2A.Z eviction is dependent on RNA polymerase II (Pol II) bearing Serine-5 phosphorylation of carboxy-terminal repeats, linking H2A.Z eviction to transcription initiation, promoter escape and early elongation" in yeast, suggesting a

general mechanism by which noncoding transcription at promoters can lead to H2A.Z erasure.

The paper is clearly written, the experiments appear to be well performed, and the results are intelligently and clearly presented. A few concerns are cited below, but in my opinion this paper is suited for publication once the points are addressed or an additional experiment is performed using the authors established methods.

1) In Figure 1, the results of tracking >1000 molecules as 2D projections are presented as histograms. Unfortunately, the expected two populations, chromatin-bound and unbound, are not cleanly resolved. Nonetheless, the peak of diffusion coefficients is best fit computationally by a model that is comprised of two populations (chromatin-bound and unbound). This is not completely satisfying, but I do point out that the authors go on to makes a fairly compelling case that their modeling is correct. First, videos are provided to allow the reader to appreciate the high quality of the single-particle tracking. Second, the authors argue that the amount of H2AZ bound is consistent with previous estimates by in vivo crosslinking, and D values are consistent with previous SPT measurements of H2B. The authors could calculate the diffusion coefficients expected based on size of the free H2A.Z and SWR1 complex and perhaps estimate the values for bound forms based on binding constants. In any case, some additional discussion of this limitation of the resolution is warranted, and perhaps some acknowledgement that there could be additional states that are also not resolved. This should require only minor editing.

2) The amount of mRNA accumulation does not correlate H2A.Z eviction, so the authors evoke the idea that early elongation in terms of short non-coding RNAs at yeast promoters may be responsible for the eviction. Can the authors take existing estimates of this short transcription genome-wide and assess if it correlates with H2A.Z eviction?

3) Does the eviction of H2A.Z happen in the presence of inhibitors of transcription that act further downstream of Cdk7, for example of the yeast equivalent of Cdk9 (Bur1)? This could serve to pinpoint the eviction at very early steps in the transcription cycle. (These inhibitors may not stop Pol II elongation, but are likely to decrease elongation rates given recently published results in Pombe (Booth et al. 2018 PMID:29416031).

4) In the Discussion, the authors state "At metazoan enhancers and promoters, turnover of H2A.Z enriched in +1 nucleosomes may be similarly coupled to pervasive ncRNA transcription likely after release from sites of Pol II pausing (Tome et al., 2018)." However, there may not be a lot of ncRNA transcription in paused regions of metazoans, as many estimates of paused Pol II half-life are relatively long and there is not much actual ncRNA transcription that goes through the first nucleosome not destined for pre-mRNA production. Perhaps the presence of a nearby transcriptionally-engaged Pol II (promoter-proximal paused Pol II) is sufficient to lead to the destabilization of the adjacent nucleosome – especially considering there is some forward motion, backtracking and TFIIS RNA realignment cleavage taking place (Nechaev et al. 2010, PMID:20007866). Also, there are factors associated with Pol II at this stage that might somehow stimulate eviction activity.

---

## [Author Response]

[Editors’ note: the authors resubmitted a revised version of the paper for consideration. What follows is the authors’ response to the first round of review.]

Reviewer #1:[…] Overall the studies are technically of high quality and the experiments are carried out in a well-controlled manner. The main conclusion that H2AZ eviction depends on both Pol II and its Ser5 phosphorylated state would be of interest to the chromatin community. However, as is, the work does not go substantially beyond the insights obtained from Tramantano et al. At the same time, the hypothesis of non-coding RNA transcription driving H2AZ turnover is attractive and novel. If the authors could provide some test of this hypothesis, this would substantially raise the significance of the work.

We thank the reviewer for appreciating the application of direct single molecule imaging to study H2A.Z dynamics in living yeast cells. To raise the significance of the work and pass the bar for *eLife* publication as noted by the senior editor, we have performed additional experiments to identify where in the transcription cycle H2A.Z is evicted. Accordingly, besides Kin28/Cdk7 and Ctk1/Cdk12, we have tested whether H2A.Z eviction is inhibited by 3 additional enzymes acting at different transcription stages (Bur1/Cdk9 kinase for Pol II, Cet1 RNA capping enzyme, and Rrp6 the 3’-5’ exonuclease for noncoding RNA degradation). We have found that among the 5 candidates, only Kin28 affects removal of H2A.Z. Thus, H2A.Z eviction is independent of RNA modifying and metabolizing activities just downstream of CTD Ser5 phosphorylation. Our findings considerably narrow the relevant stage of the transcription cycle where histone eviction occurs to early Pol II elongation after promoter escape, but not to productive elongation. We believe that these additional experiments provide sufficient new mechanistic insight to the study. (Testing the hypothesis of non-coding RNA transcription driving H2AZ turnover is an important issue, but beyond the scope of the current work. As noted below in our response to reviewer #3, point #2, a bioinformatic approach is currently unfeasible for technical reasons).

Reviewer #2:The manuscript described the role of Pol2 in H2A.Z dynamics by using single-molecule tracking. Authors measure H2A.Z bound fractions in various depletion conditions, concluding that H2A.Z eviction is dependent on Pol2-CTD serine-5p rather than INO80 complex. Although single-molecule tracking for H2A.Z in the various conditions is informative to the field, the biological significance seems to be a large repetition of the results in Tramantano, 2016, with different aspects of technique. It would be nice if authors could more clearly explain the novelty of their work and how it extends beyond Tramantano, 2016.

We concur and new experiments in the revised manuscript addresses the reviewer’s point. Copied below is our response to the same concern from reviewer #1.

To raise the significance of the work and pass the bar for *eLife* publication as noted by the senior editor, we have performed additional experiments to identify where in the transcription cycle H2A.Z is evicted. Accordingly, besides Kin28/Cdk7 and Ctk1/Cdk12, we have tested whether H2A.Z eviction is inhibited by 3 additional enzymes acting at different transcription stages (Bur1/Cdk9 kinase for Pol II, Cet1 RNA capping enzyme, and Rrp6 the 3’-5’ exonuclease for noncoding RNA degradation). We have found that among the 5 candidates, only Kin28 affects removal of H2A.Z. Thus, H2A.Z eviction is independent of RNA modifying and metabolizing activities just downstream of CTD Ser5 phosphorylation. Our findings considerably narrow the relevant stage of the transcription cycle where histone eviction occurs to early Pol II elongation after promoter escape, but not to productive elongation. We believe that these additional experiments provide sufficient new mechanistic insight to the study.

We think the Tramantano, 2016 paper is pretty tightly controlled and well-executed, even though its findings are largely correlative. Their correlations tell them that both Pol II and PIC are important for H2A.Z turnover at the +1 nucleosome, but the mechanism is unknown and which part of the PIC is necessary is unclear. So the major contribution from this paper is that a defective TFIIH can restore H2A.Z fraction bound to near WT levels in a Swc5-depleted background. Going back to the paper from which the Wu lab got the Kin28is mutant puts a question mark on their proposed mechanism of H2A.Z turnover. They found that in Kin28is cells treated with CMK, Pol II accumulates at the +2 nucleosome. So the Wu hypothesis that it is passage of Pol II through the +1 nucleosome that displaces H2A.Z doesn't make sense. It would be nice to have the authors comment on this and help clarifying their point.

We appreciate the reviewer’s comment on interpretation of the effects of the Kin28is mutant on Pol II. We also wrestled with the published model for the Kin28is mutant, and recognized that the presented model highlighting Pol II accumulation between +1 and +2 nucleosomes after CMK inhibition of Kin28 kinase does not entirely reflect the actual data in the publication, which shows Pol II accumulation in a broader region at the promoter from approx. -150 to + 170 (Figure 5A, C) (Rodriguez-Molina et al., 2016)). Irrespective, we have performed a further control analyzing effects of CMK-treated Kin28is without Swc5-AA. This revealed that Kin28is actually affects H2A.Z deposition (reason unclear), thus limiting the utility of this mutant for testing H2A.Z eviction. Hence, we used an alternative Kin28-AA mutant in a double Kin28-AA; Swc5-AA experiment to unveil the eviction pathway in the new experiment shown in Figure 4. (Importantly, the single Kin28-AA depletion did not adversely affect H2A.Z deposition (Figure 4—figure supplement 1A-C). We have removed the original data, replacing it with new findings using the Kin28-AA strain (Figure 4).

-The key conclusion in the manuscript is that Pol2-CTD-S5p associates with H2A.Z eviction based on the kin28is model. However, it is problematic to disrupt the TFIIH helicase activity for this purpose, in which this would have nothing to do with phosphorylation but shutdown entire transcription.

We thank the reviewer for drawing our attention to the possibility that Kin28 inactivation or now, depletion, could perturb the helicase activity of TFIIH. Accordingly, the following sentence is added in the revised Discussion: “A role for Kin28 in Ssl2-facilitated TSS scanning by Pol II is not excluded for H2A.Z eviction, although it has been shown that depletion of Kin28 by AA does not alter TSS usage (Murakami et al., 2015).”

-Authors used the time course with dye staining before anchoring protein away. This resulted in the possibility that a portion of SPT measurements might be derived from the "WT" condition. Could authors also demonstrate a time course with dye staining after anchoring target away, or staining with different dyes before and after adding rapamycin?

All SPT data was acquired after one hour of cell growth in presence of rapamycin. This is the standard AA protocol in the field under which maximum depletion is achieved. To be explicit, we have inserted this sentence in the Figure 2 legend: ‘Rapamycin treatment for an hour before SPT, and imaging was performed in continued presence of rapamycin’.

-Authors did not have any data supporting the role of ncRNA here. Authors need to provide the evidence by measuring H2A.Z dynamics in RRP6 or XRN1 mutants.

As stated in the response above, we have performed the requested experiment and present new data examining role of Rrp6 on H2A.Z eviction (Figure 4—figure supplement 1).

Reviewer #3:[…] The paper is clearly written, the experiments appear to be well performed, and the results are intelligently and clearly presented. A few concerns are cited below, but in my opinion this paper is suited for publication once the points are addressed or an additional experiment is performed using the authors established methods.1) In Figure 1, the results of tracking >1000 molecules as 2D projections are presented as histograms. Unfortunately, the expected two populations, chromatin-bound and unbound, are not cleanly resolved. Nonetheless, the peak of diffusion coefficients is best fit computationally by a model that is comprised of two populations (chromatin-bound and unbound). This is not completely satisfying, but I do point out that the authors go on to makes a fairly compelling case that their modeling is correct. First, videos are provided to allow the reader to appreciate the high quality of the single-particle tracking. Second, the authors argue that the amount of H2AZ bound is consistent with previous estimates by in vivo crosslinking, and D values are consistent with previous SPT measurements of H2B. The authors could calculate the diffusion coefficients expected based on size of the free H2A.Z and SWR1 complex and perhaps estimate the values for bound forms based on binding constants. In any case, some additional discussion of this limitation of the resolution is warranted, and perhaps some acknowledgement that there could be additional states that are also not resolved. This should require only minor editing.

We thank the reviewer for his thoughtful comments, and wish to point out that the quality of population histograms are in line with publications in the field. We have revised our text to emphasize that we have chosen the simplest model to fit data and do not exclude the presence of additional minor populations with distinct diffusive values. Regarding the values for free diffusion, because of constraints of motion blurring and focal depth and nuclear membrane confinements, it is challenging to precisely measure the D for free molecules. Nonetheless, it is useful to note that the average D for free H2A.Z (chaperoned H2A.Z-H2B dimer, 1.17 μm^2^s^-1^) and D for free Swr1 (Swr1 complex, 0.62 μm^2^s^-1^) reflects their different molecular sizes.

2) The amount of mRNA accumulation does not correlate H2A.Z eviction, so the authors evoke the idea that early elongation in terms of short non-coding RNAs at yeast promoters may be responsible for the eviction. Can the authors take existing estimates of this short transcription genome-wide and assess if it correlates with H2A.Z eviction?

The eviction rate of H2A.Z at yeast promoters can be estimated, in theory, based on the depletion rate of H2A.Z upon Swc5 depletion using the ChIP-seq data from Tramantano et al., 2016. However, since different promoters have different initial steady-state levels of H2A.Z prior to Swc5 perturbation, the slope of H2A.Z depletion is influenced by prior H2A.Z deposition, preventing a direct measurement of eviction. To our knowledge, there is currently no good genome-wide dataset for the rates of H2A.Z eviction. Therefore, at this time, we cannot confidently compare noncoding transcription with H2A.Z eviction.

3) Does the eviction of H2A.Z happen in the presence of inhibitors of transcription that act further downstream of Cdk7, for example of the yeast equivalent of Cdk9 (Bur1)? This could serve to pinpoint the eviction at very early steps in the transcription cycle. (These inhibitors may not stop Pol II elongation, but are likely to decrease elongation rates given recently published results in Pombe (Booth et al. 2018 PMID:29416031).

We have imaged H2A.Z in cells with double AA of Bur1/Cdk9 along with Swc5. We find, that unlike Kin28, depletion of Bur1 does not inhibit H2A.Z eviction (Figure 4). As noted above, we also examined whether H2A.Z eviction is inhibited by Cet1 RNA capping enzyme, and Rrp6 the 3’-5’ exonuclease for noncoding RNA degradation, and found them lacking effects on H2A.Z eviction. Thus, H2A.Z eviction is independent of RNA modifiying and metabolizing activities just downstream of CTD Ser5 phosphorylation.

4) In the Discussion, the authors state "At metazoan enhancers and promoters, turnover of H2A.Z enriched in +1 nucleosomes may be similarly coupled to pervasive ncRNA transcription likely after release from sites of Pol II pausing (Tome et al., 2018)." However, there may not be a lot of ncRNA transcription in paused regions of metazoans, as many estimates of paused Pol II half-life are relatively long and there is not much actual ncRNA transcription that goes through the first nucleosome not destined for pre-mRNA production. Perhaps the presence of a nearby transcriptionally-engaged Pol II (promoter-proximal paused Pol II) is sufficient to lead to the destabilization of the adjacent nucleosome – especially considering there is some forward motion, backtracking and TFIIS RNA realignment cleavage taking place (Nechaev et al. 2010, PMID:20007866). Also, there are factors associated with Pol II at this stage that might somehow stimulate eviction activity.

We take note of reviewer’s comment and have rephrased our Discussion to accommodate potential effects of paused Pol II on H2A.Z eviction. The revised sentence reads “At metazoan promoters, turnover of H2A.Z enriched in +1 nucleosomes may be similarly coupled to transcription in the process of Pol II pausing and release (Tome et al., 2018). “